# Development and validation of a model for predicting incident type 2 diabetes using quantitative clinical data and a Bayesian logistic model: A nationwide cohort and modeling study

Lua Wilkinson[1,2]*, Nengjun Yi[3], Tapan Mehta[4], Suzanne Judd[3], W. Timothy Garvey[1,5]

**1** Department of Nutrition Sciences, University of Alabama at Birmingham, Birmingham, Alabama, United States of America, **2** Novo Nordisk, Plainsboro, New Jersey, United States of America, **3** Department of Biostatistics, University of Alabama at Birmingham, Birmingham, Alabama, United States of America, **4** Department of Health Services Administration, University of Alabama at Birmingham, Birmingham, Alabama, United States of America, **5** Birmingham VA Medical Center, Alabama, United States of America

\* Lua.wilkinson@gmail.com

**Data Availability Statement:** This study uses data from the Reasons for Geographic and Racial Differences in Stroke (REGARDS) cohort. In order

## Abstract

### Background

Obesity is closely related to the development of insulin resistance and type 2 diabetes (T2D). The prevention of T2D has become imperative to stem the rising rates of this disease. Weight loss is highly effective in preventing T2D; however, the at-risk pool is large, and a clinically meaningful metric for risk stratification to guide interventions remains a challenge. The objective of this study is to predict T2D risk using full-information continuous analysis of nationally sampled data from white and black American adults age ≥45 years.

### Methods and findings

A sample of 12,043 black (33%) and white individuals from a population-based cohort, REasons for Geographic And Racial Differences in Stroke (REGARDS) (enrolled 2003–2007), was observed through 2013–2016. The mean participant age was 63.12 ± 8.62 years, and 43.7% were male. Mean BMI was 28.55 ± 5.61 kg/m$^2$. Risk factors for T2D regularly recorded in the primary care setting were used to evaluate future T2D risk using Bayesian logistic regression. External validation was performed using 9,710 participants (19% black) from Atherosclerotic Risk in Communities (ARIC) (enrolled 1987–1989), observed through 1996–1998. The mean participant age in this cohort was 53.86 ± 5.65 years, and 44.6% were male. Mean BMI was 27.15 ± 4.92 kg/m$^2$. Predictive performance was assessed using the receiver operating characteristic (ROC) curves and area under the curve (AUC) statistics. The primary outcome was incident T2D. By 2016 in REGARDS, there were 1,602 incident cases of T2D. Risk factors used to predict T2D progression included age, sex, race, BMI, triglycerides, high-density lipoprotein, blood pressure, and blood glucose. The

to abide by its obligations with NIH/NINDS and the Institutional Review Board of the University of Alabama at Birmingham, REGARDS facilitates data sharing through formal data use agreements. Any investigator is welcome to access the REGARDS data through this process. Requests for data access may be sent to regardsadmin@uab.edu. For R codes, readers can contact Nengjun Yi, nyi@uab. edu.

**Funding:** This research project is supported by cooperative agreement U01 NS041588 co-funded by the National Institute of Neurological Disorders and Stroke (NINDS) and the National Institute on Aging (NIA), National Institutes of Health, Department of Health and Human Service. The content is solely the responsibility of the authors and does not necessarily represent the official views of the NINDS or the NIA. Representatives of the NINDS were involved in the review of the manuscript but not directly involved in the collection, management, analysis or interpretation of the data. The authors thank the other investigators, the staff, and the participants of the REGARDS study for their valuable contributions. A full list of participating REGARDS investigators and institutions can be found at http://www. regardsstudy.org. Additionally, the authors acknowledge support from the UAB Obesity Training Program (T32 DK062710); the American Heart Association Strategically Focused Obesity Research Network center at the University of Alabama at Birmingham (17SFRN33610070); the Merit Review program of the Department of Veterans Affairs (I01CX000432); the UAB Diabetes Research Center (P30 DK079626); and the UAB Nutrition Obesity Research Center (DK056336). The funders had no role in study design, data collection and analysis, decision to publish, or preparation of the manuscript.

**Competing interests:** There has been no financial support to any author for this manuscript, and the opinions expressed are entirely those of the authors. L.W. is an employee of Novo Nordisk Inc. W.T.G has served on ad hoc advisory boards for Sanofi, Novo Nordisk, Gilead, Boehringer-Ingelheim, Jazz Pharmaceuticals, BOYDSense, and the American Medical Group Association, and he has conducted research sponsored by the University of Alabama at Birmingham funded by Sanofi, Merck/Pfizer, and Novo Nordisk.

**Abbreviations:** ARIC, Atherosclerotic Risk in Communities; ATP III, Adult Treatment Panel III; AUC, area under the curve; CC, correlation coefficient; CMDS, cardiometabolic disease staging; DBP, diastolic blood pressure; HDL, high-density lipoprotein; OR, odds ratio; REGARDS,

Bayesian logistic model (AUC = 0.79) outperformed the Framingham risk score (AUC = 0.76), the American Diabetes Association risk score (AUC = 0.64), and a cardiometabolic disease system (using Adult Treatment Panel III criteria) (AUC = 0.75). Validation in ARIC was robust (AUC = 0.85). Main limitations include the limited generalizability of the REGARDS sample to black and white, older Americans, and no time to diagnosis for T2D.

## Conclusions

Our results show that a Bayesian logistic model using full-information continuous predictors has high predictive discrimination, and can be used to quantify race- and sex-specific T2D risk, providing a new, powerful predictive tool. This tool can be used for T2D prevention efforts including weight loss therapy by allowing clinicians to target high-risk individuals in a manner that could be used to optimize outcomes.

## Author summary

### Why was this study done?

- Obesity affects approximately 42% of the US population and causes significant morbidity, including a marked increase in insulin resistance and type 2 diabetes (T2D), and varies by sex and race.

- Weight loss is effective in preventing T2D, but the at-risk pool is large and weight loss interventions are time-consuming and costly.

- A simple tool to identify those at risk for developing T2D is needed.

### What did the researchers do and find?

- We performed a Bayesian logistic regression study using data from the national REGARDS (2003–2016) and ARIC (1987–1998) cohorts in the United States for risks associated with T2D in black and white American adults.

- We investigated 8 demographic and metabolic syndrome risk factors for T2D and incorporated Bayesian hierarchical techniques into the development of a risk prediction calculation.

- These 8 simple traits showed improved ability to predict progression to T2D compared to other commonly used paradigms, and can be used in clinical settings to target those at high risk for developing T2D.

### What do these findings mean?

- Using a different methodology, with simple, objective traits regularly measured in a clinical setting (by tests that can be performed by non-specialists), we showed that

REasons for Geographic And Racial Differences in
Stroke; ROC, receiver operating characteristic;
SBP, systolic blood pressure; T2D, type 2 diabetes.

metabolic traits related to insulin resistance can be used to predict T2D in black and
white American adults.

- Rational strategies such as this can be used by clinicians to quantitatively assess T2D
  risk among those with obesity at high risk for the disease.

## Introduction

The prevalence of type 2 diabetes (T2D) continues to rise, creating a greater burden in patients
and adverse impacts in public health [1]. The rising prevalence of T2D is linked to escalating
rates of obesity, and both T2D and obesity disproportionately affect certain populations, often
along social, demographic, or economic lines. For example, non-Hispanic black Americans
are affected by T2D with almost double the prevalence (13.4%) of non-Hispanic whites (7.3%),
and also exhibit higher rates of obesity—particularly when comparing black and white women
(13.2% versus 6.8%, respectively) [1].

Strategies for effective T2D prevention have become critically important to reduce the
impact of this disease. A robust body of evidence is conclusive that weight loss is highly effec-
tive in preventing T2D—regardless of whether weight loss is achieved through lifestyle therapy
[2], anti-obesity medications [3], or bariatric surgery [4]. However, the challenge that remains
is 2-fold: First, sustained weight loss using the current tools of obesity management are labor
intensive on the part of both the healthcare team and the patient, and, second, the at-risk pool
of patients for T2D is quite large. By way of illustration, the National Health and Nutrition
Examination Survey (NHANES) demonstrated that, in 2013–2014, 70.7% of US adults had
overweight or obesity and 34.2% had metabolic syndrome, and all these individuals are at high
risk of developing T2D [1].

Clearly, risk stratification approaches are needed to identify those at highest risk of T2D,
and to optimize the benefit/risk ratio and cost-effectiveness of the application of weight loss
therapy in the prevention of T2D. The majority of risk assessment strategies use binary predic-
tors for risk factors, including those employed by National Cholesterol Education Program
Adult Treatment Panel III (ATP III) [5]. Discretizing continuous predictors can result in the
loss of valuable information and reduce the clinical usefulness of the predictive model [6]. For
example, the risk conferred by metabolic syndrome traits exists over a spectrum of values, and
binary responses do not adequately classify T2D risk over the quantitative range of risk factors
[7]. Finally, the predictive value of various risk factors and risk scores may not be generalizable
from one population to another. In particular, African Americans have been understudied
with respect to risk models, score development, replication, and validation [8].

Guo et al. [9] earlier developed a cardiometabolic disease staging (CMDS) system using
binary predictors using data from the Coronary Artery Risk Development in Young Adults
(CARDIA) [10] and Atherosclerotic Risk in Communities (ARIC) [11] cohorts to predict inci-
dent T2D with specificity for sex and race. CMDS was developed using quantitative measures
of metabolic syndrome traits (i.e., ATP III criteria) [12], with the limitation that these cohorts
were not designed as nationally representative. Additionally, a binary prediction approach
such as this does not fully take into account the risk conferred by cardiometabolic disease
manifestations due to pathophysiological processes of adipocyte dysfunction, systemic inflam-
mation, and oxidative stress [13]. There have been attempts to observe an association between
metabolic syndrome $z$-scores and risk of future T2D using a continuous metabolic severity

score [14]. However, these analyses fitted separate logistic models for each metabolic syndrome trait and did not consider possible interactions, such as between high-density lipoprotein (HDL) and triglycerides [15].

Our current objective was to create a highly predictive score that rigorously captures race and sex differences in T2D risk. This was done using a large national cohort of black and white Americans from the REasons for Geographic And Racial Differences in Stroke (REGARDS) study. We compared the predictive ability of a CMDS T2D prediction model using individual laboratory and anthropometric measurements as continuous functions with Bayesian logistic regression. We also compared the predictive accuracy of enhanced CMDS with other existing T2D prediction scores by looking at receiver operating characteristic (ROC) curves and area under the curve (AUC) statistics. The purpose of this analysis is to create a tool using quantitative predictors available in real-world clinical practice that identifies individuals who are most likely to benefit from therapies to prevent T2D. The application of CMDS allows clinicians treating those with overweight/obesity to target effective weight loss strategies in those at highest risk of T2D, in order to optimize the benefit/risk ratio and cost-effectiveness of interventions.

## Methods

The institutional review board of the University of Alabama at Birmingham designated this analysis as not human subjects research and waived the need for approval. The analyses were prespecified and approved by the REGARDS committee. This study is reported as per the Transparent Reporting of a Multivariable Prediction Model for Individual Prognosis or Diagnosis guideline (S1 TRIPOD Checklist).

### Study populations

The enhanced CMDS model was developed in REGARDS and externally validated in ARIC. REGARDS was chosen as it is one of the largest and most recent surveys of black and white adults that collected information relevant to T2D risk. ARIC was chosen for external validation as it is a recent longitudinal scientific sample of black and white Americans. These analyses used only de-identified data.

**REGARDS.** The REGARDS study is an ongoing longitudinal survey designed to look at stroke mortality of black and white Americans. The design has been reported elsewhere [16]. In this scientific sample from the US, a total of 30,239 black and white men and women age 45 years and older from 48 states and the District of Columbia were enrolled between 2003 and 2007. Participants were interviewed by telephone, followed by an in-home visit for physiological measures and obtaining biosamples, at baseline, and then observed for a median follow-up duration of 9.5 ± 0.9 years (second in-home visit, 2013–2016). Follow-up time is rounded to 10 years for reporting. Information on incident T2D was collected at baseline and follow-up. We restricted the analysis to those without T2D at baseline who had completed the second in-home visit. Between the first and second visit, 5,713 individuals died, and 8,532 withdrew from further follow-up, leaving a population of 15,938 with follow-up data available. Individuals with T2D at baseline ($n = 3,260$) and those missing relevant covariate information at baseline ($n = 635$) were excluded, leaving a final study population of 12,043 individuals. Site institutional review boards approved the protocol, and informed consent was obtained.

Collection of blood specimens, physical measurements, and urine was performed using standardized methods. Participants were asked to fast for 10–12 hours before the visit ($n = 9,332$). Those who did not fast ($n = 1,440$) or had no information on fasting ($n = 1,271$) were included as non-fasters. T2D was defined by fasting blood glucose level $\geq 7.0$ mmol/l,

non-fasting blood glucose $\geq$ 11.1 mmol/l, self-reported T2D, or being on diabetes medication. Race was defined by self-report as black or white. Standardized blood pressure was taken twice in-home and calculated as the average of the 2 measurements. Lipids were assayed using either the fasting or non-fasting sample.

**ARIC.** The ARIC study is a longitudinal, ongoing prospective study initiated in 1987 [17]. ARIC includes 15,792 black and white men and women age 45–64 years at baseline from 4 US communities: Jackson, Mississippi; Forsyth County, North Carolina; Minneapolis, Minnesota; and Washington County, Maryland. Individuals were interviewed at 4 distinct follow-up time points between 1990 and 2013. We restricted this analysis to 2 time points (1987–1989 and 1996–1998), matching the length of follow-up in REGARDS. Information on T2D was collected at both time points—those with T2D at baseline and/or missing relevant covariate information were excluded, along with those lost to follow-up or death; the final population included 9,710 individuals.

Analysis of fasting and plasma specimens was performed at central laboratories. For incident T2D, we included those with self-report of T2D or being on T2D medication, as described by the ARIC protocol [18]. Site institutional review boards approved the study at each site, and informed consent was obtained.

## Predictors used to determine T2D risk

In order to predict future T2D we relied on objective, quantitative traits commonly available in clinical care venues, particularly in patients presenting with obesity or metabolic syndrome: blood glucose, BMI and waist circumference, systolic blood pressure (SBP) and diastolic blood pressure (DBP), HDL cholesterol, and triglycerides [19]. We assessed these traits as continuous predictors. Additionally, in order to improve clinical and general applications of the score, we examined, using correlation matrices and AUC, whether substituting BMI for waist circumference changed predictive ability.

## Statistical methods

We used Bayesian logistic regression models to analyze our data by jointly fitting prespecified predictors and/or their interactions. Following Gelman et al. [20], we used weakly informative Cauchy priors, which have the advantage of providing minimal prior information to constrain the coefficients in a reasonable range, stabilizing the model fitting, and improving the model prediction performance [20,21]. We fitted the Bayesian logistic regression models with Cauchy priors by incorporating an approximate expectation-maximization algorithm into the usual iteratively weighted least squares used in classical logistic regression. For large datasets and only a few predictors, conventional logistic regression may perform similarly to Bayesian logistic regression. However, Bayesian models with weakly informative priors can provide more reliable results if there are problems of correlation and overfitting.

We built a Bayesian logistic model using REGARDS and evaluated its predictive values in ARIC. We used several measures to assess the predictive performance, including AUC, mean squared error, and misclassification [22,23]. We compared the main-effect model, with only the main effects of the predictors mentioned above, with the interacting model, which included all the main effects and also multiple interactions, including sex × race, SBP × DBP, HDL × triglycerides, waist circumference × BMI, BMI × HDL, and BMI × triglycerides. We also tested if using mean arterial pressure conferred any benefit over SBP and DBP. The model fitting and predictive evaluation were implemented using R package BhGLM (Bayesian hierarchical generalized linear models) (https://github.com/nyiuab/BhGLM) [24].

We also compared our method with several other predictive modeling methods, including lasso, generalized additive modeling, random forests, and support vector machine learning (S1 Text). We found that our Bayesian logistic model outperformed these alternative approaches (S1 Table).

To create a useable, interactive instrument, we calculated the predictive risk probabilities based on the fitted Bayesian logistic model, allowing one to simply input an individual's actual data into a computer program and receive a risk probability based on his/her personal anthropometric, demographic, and laboratory values. The formula for calculating the predictive risk probabilities of incident T2D can be found in S2 Text.

## Comparisons to other risk scores

We compared the AUC from the Bayesian logistic model with the CMDS model [19] developed using discontinuous traits conforming with ATP III criteria. We also report the differences in AUC between the current Bayesian logistic model and the Framingham [25] and American Diabetes Association [26] risk scores. We recalculated the AUC for these scores using logistic regression methods and available REGARDS data; we were unable to include family history, as it is a somewhat subjective and nonquantitative variable that is unavailable for REGARDS participants.

## Results

Baseline characteristics of study participants are reported in Table 1. In REGARDS, 12,043 eligible participants without T2D at baseline completed the follow-up examination and had

**Table 1. Baseline characteristics of included participants.**

| Characteristic | Study | | | | | |
|---|---|---|---|---|---|---|
| | REGARDS | | | | | ARIC |
| | Total | Black women | Black men | White women | White men | Total |
| Population $n$ | 12,043 | 2,578 | 1,394 | 4,204 | 3,867 | 9,710 |
| White[1], $n$ (%) | 8,071 (67) | | | | | 7,906 (81.4) |
| Male[1], $n$ (%) | 5,261 (43.7) | | | | | 4,326 (44.6) |
| Age (years) | 63.12 (8.62) | 62.19 (8.69) | 62.27 (8.24) | 63.18 (8.81) | 63.98 (8.43) | 53.86 (5.65) |
| Body mass index (kg/m$^2$) | 28.55 (5.61) | 31.16 (6.67) | 28.63 (5.07) | 27.56 (5.65) | 27.87 (4.27) | 27.15 (4.92) |
| Waist circumference (cm) | 93.24 (14.3) | 93.94 (14.32) | 97.71 (12.80) | 86.50 (13.97) | 98.48 (12.02) | 95.57 (13.12) |
| Systolic blood pressure (mm Hg) | 125.00 (15.49) | 127.51 (16.43) | 129.21 (15.37) | 121.65 (15.35) | 125.45 (14.27) | 118.66 (16.99) |
| Diastolic blood pressure (mm Hg) | 76.33 (9.18) | 77.99 (9.34) | 79.15 (9.52) | 74.14 (8.79) | 76.60 (8.81) | 72.88 (10.58) |
| Blood glucose (mmol/l) | 5.16 (0.68) | 5.21 (0.74) | 5.27 (0.74) | 5.05 (0.62) | 5.19 (0.67) | 5.47 (0.51) |
| HDL cholesterol (mmol/l) | 1.38 (0.42) | 1.52 (0.41) | 1.26 (0.37) | 1.54 (0.42) | 1.17 (0.34) | 1.36 (0.44) |
| Triglycerides (mmol/l) | 1.42 (0.89) | 1.15 (0.61) | 1.31 (1.21) | 1.48 (0.78) | 1.56 (0.98) | 1.40 (0.85) |
| ATP III[2], $n$ (%) | | | | | | |
| 0 risk factors | 2,113 (17.5) | 254 (9.9) | 199 (14.3) | 976 (23.2) | 684 (17.7) | 1,666 (17.2) |
| 1 risk factor | 3,340 (27.7) | 603 (23.4) | 462 (33.1) | 1,143 (27.2) | 1,132 (29.3) | 2,521 (26.0) |
| 2 risk factors | 3,244 (26.9) | 881 (34.2) | 415 (29.8) | 984 (23.4) | 964 (24.9) | 2,336 (24.1) |
| 3 or more risk factors | 2,246 (27.8) | 840 (32.6) | 318 (22.8) | 1,101 (26.2) | 1,087 (28.1) | 3,189 (32.8) |
| Diabetes incidence[3] at second in-home visit, $n$ (%) | 1,602 (13.3) | 482 (18.7) | 257 (18.4) | 386 (9.2) | 477 (12.3) | 927 (9.5) |

Data are mean (SD) unless otherwise indicated.

[1]Race and sex were self-reported.

[2]Risk factors defined as follows: fasting blood glucose > 5.55 mmol/l; waist circumference > 102 cm in men, >88 cm in women; systolic blood pressure > 130 mm Hg or diastolic blood pressure > 85 mm Hg or on antihypertensive medication; HDL cholesterol < 1.03 mmol/l in men, <1.29 mmol/l in women; and fasting triglycerides > 1.69 mmol/l or on lipid-lowering medication.

[3]Incident diabetes is defined as fasting glucose ≥ 7.0 mmol/l, non-fasting glucose ≥ 11.1 mmol/l, currently on medication for diabetes, or self-report of diabetes diagnosis.

ARIC, Atherosclerotic Risk in Communities; ATP III, Adult Treatment Panel III; HDL, high-density lipoprotein; REGARDS, REasons for Geographic And Racial Differences in Stroke.

complete data on relevant covariates (mean age 63.1 years, range 45–92 years; 33% black). During a follow-up time ranging from 7.4 to 13.4 years (median 9.5 ± 0.9 years), there were 1,602 cases of new T2D (13.3%). Approximately 75% of participants were overweight or had obesity. Ranges of all variables included are reported in S2 Table.

For external validation using ARIC, 9,710 participants completed the follow-up examination (mean age 53.9 years, range 45–66 years; 19% black) and had complete data on relevant covariates. During a follow-up time of 10 years, there were 927 cases of new T2D (9.5%). Almost 65% of participants had overweight or obesity.

Black females had the highest incidence of T2D in REGARDS (18.7%) and ARIC (16.3%); white females had the lowest (9.2% and 6.4%). Black females had the highest prevalence of obesity in both surveys, using both BMI (51% and 43%) and elevated waist circumference (64% and 74%). In terms of cardiometabolic risk profile, 34% of people in REGARDS presented with metabolic syndrome, 33% in ARIC.

### The fitted models and their predictive values

Results from the fitted Bayesian logistic model with main effects are presented in Table 2. The model fitted in REGARDS had an AUC of 0.79 (95% CI 0.78–0.80). External validation using the model generated in REGARDS was conducted in the ARIC cohort, for which the AUC was 0.85 (95% CI 0.83–0.86). This model included the variables of SBP, DBP, blood glucose, BMI, HDL, triglycerides, age (45–92 years), sex (male or female), and race (black or white). Importantly, this model incorporated risk conferred over the continuum of values for each risk factor as well as the effect that age, race, and sex have on the contributions of the factors to overall T2D risk.

**Table 2. Predictive power, validation, and interactions.**

| Model | AUC | MSE[1] | Misclassification |
|---|---|---|---|
| REGARDS: Development[2] | 0.789 | 0.099 | 0.131 |
| ARIC: External validation | 0.846 | 0.074 | 0.090 |
| **Interactions[3]** | | | |
| Sex and race with main effects[4] | 0.794 | 0.098 | 0.130 |
| SBP × DBP[5] | 0.788 | 0.099 | 0.131 |
| MAP[6] | 0.789 | 0.099 | 0.131 |
| HDL × triglycerides[7] | 0.779 | 0.100 | 0.133 |
| Waist circumference × BMI[8] | 0.780 | 0.100 | 0.132 |
| BMI × HDL[9] | 0.787 | 0.099 | 0.131 |
| BMI × triglycerides[10] | 0.785 | 0.100 | 0.132 |

[1]MAP calculated as the average squared difference between the observed and predicted values.

[2]Analyzed using Bayesian logistic regression. Diabetes incidence ~ age + sex + race + BMI + triglycerides + HDL cholesterol + SBP + DBP + blood glucose.

[3]Analyzed using the REGARDS dataset by Bayesian logistic regression.

[4]Diabetes incidence ~ (age + BMI + triglycerides + HDL cholesterol + SBP + DBP + blood glucose) × (sex:race).

[5]Diabetes incidence ~ age + sex + race + BMI + triglycerides + HDL cholesterol + SBP:DBP + blood glucose.

[6]MAP calculated as [(2 ×DBP) + SBP]/3. Diabetes incidence ~ age + sex + race + BMI + triglycerides + HDL cholesterol + MAP + blood glucose.

[7]Diabetes incidence ~ age + sex + race + BMI + triglycerides:HDL cholesterol + SBP + DBP + blood glucose.

[8]Diabetes incidence ~ age + sex + race + BMI:waist circumference + triglycerides + HDL cholesterol + SBP + DBP + blood glucose.

[9]Diabetes incidence ~ age + sex + race + BMI:HDL cholesterol + triglycerides + SBP + DBP + blood glucose.

[10]Diabetes incidence ~ age + sex + race + BMI:triglycerides + HDL cholesterol + SBP + DBP + blood glucose.

ARIC, Atherosclerotic Risk in Communities; AUC, area under the curve; DBP, diastolic blood pressure; HDL, high-density lipoprotein; MAP, mean arterial pressure; MSE, mean squared error; REGARDS, REasons for Geographic And Racial Differences in Stroke; SBP, systolic blood pressure.

We did not observe significant improvements in AUC and other measures when including interactions in the predictive model, and, in fact, only observed a mild improvement when interacting sex and race with main effects, where the AUC went from 0.789 (main effects with sex and race included as main effects, no interaction) to 0.794 (interaction). Inclusion of interactions involving SBP and DBP, HDL and triglycerides, and waist circumference and BMI did not enhance the predictive accuracy, nor did the substitution of mean arterial pressure for DBP and SBP.

Fig 1 shows the estimated odds ratios (ORs) of incident T2D for individual risk factors in REGARDS based on the main-effect model. All factors used to construct the fitted model, except for sex and DBP, significantly impact the risk of T2D. While these other factors significantly provide additional information about T2D risk when added to the model, the risk factors associated with the greatest impact on odds of future T2D were blood glucose (OR 1.06, 95% CI 1.06 to 1.07) and race (white, OR 0.63, 95% CI 0.56 to 0.71).

## Correlation of parameters: Waist circumference and BMI

Waist circumference and BMI are similarly correlated with T2D risk (waist circumference correlation coefficient [CC] 0.19; BMI CC 0.19). Additionally, waist circumference and BMI are correlated with each other (CC 0.74). Although waist circumference is not routinely assessed in many clinical venues, we alternatively analyzed its predictive power, and found that the AUC for the model with waist circumference was 0.791, compared to 0.789 for the model with BMI. While waist circumference did appear to confer a minimal improvement to the AUC, given the clinical application of the Bayesian logistic model, BMI is an appropriate substitute for waist circumference without substantial loss of predictability. In addition, we tested the correlation between all other parameters: SBP and DBP showed a CC of 0.62; all other pairs of parameters had CC < 0.2.

## Comparisons to previous models

Fig 2 presents ROC curves comparing the Bayesian logistic model using continuous variables to our previous score model using binary ATP III criteria and age predictors (AUC 0.75) [19],

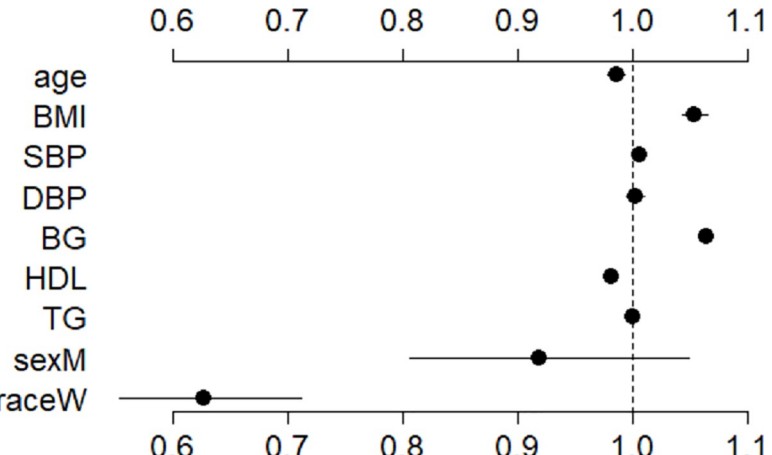

**Fig 1. Odds ratios of incident type 2 diabetes for individual risk factors used to construct the fitted main-effect logistic model.** The points and lines present the estimated values and 95% CIs, respectively. Odds ratios are as follows: systolic blood pressure (SBP), 1.006 (95% CI 1.001 to 1.011); diastolic blood pressure (DBP), 1.003 (95% CI 0.995 to 1.012); blood glucose (BG), 1.064 (95% CI 1.059 to 1.069); BMI, 1.055 (95% CI 1.044 to 1.066); high-density lipoprotein (HDL) cholesterol, 0.982 (95% CI 0.979 to 0.987); triglycerides (TG), 1.001 (95% CI 1.001 to 1.002); age, 0.987 (95% CI 0.979 to 0.994); white race (raceW), 0.628 (95% CI 0.556 to 0.709); male sex (sexM), 0.919 (95% CI 0.808 to 1.046). The references for the binary predictors race and sex are black and female, respectively.

**Comparing Scores**

**Fig 2. Receiver operating characteristic curves for the Bayesian logistic model (BhGLM), the CMDS score based on discontinuous ATP III criteria, the Framingham risk score, and the American Diabetes Association risk score in the REGARDS cohort.** The Bayesian score included the following risk factors: age, sex, race, BMI, triglycerides, HDL cholesterol, blood pressure, and blood glucose. The CMDS score using ATP III criteria thresholds included sex, race, BMI, triglycerides, HDL cholesterol, blood pressure, and blood glucose using binary ATP III criteria. The Framingham risk score was a simple clinical score using fasting glucose, BMI, HDL cholesterol, triglycerides, and blood pressure. The American Diabetes Association risk score included age, sex, blood pressure, BMI, and physical activity. ATP III, Adult Treatment Panel III; CMDS, cardiometabolic disease staging; HDL, high-density lipoprotein; REGARDS, REasons for Geographic And Racial Differences in Stroke.

as well as the Framingham (AUC 0.76) and American Diabetes Association (AUC 0.64) scoring systems using logistic regression methods with variables available in REGARDS. The AUC for the Bayesian logistic model, using continuous variables from REGARDS, was 0.79.

## Predictive risk probabilities

Based on the fitted main-effect logistic model, we obtained a formula for calculating the probability of T2D for any individual given the values of the risk factors (S2 Text). Results for the probability of T2D as predicted by each individual risk factor included in the final Bayesian logistic model are displayed in Fig 3 over the continuum of values, stratified by sex and race. There are several salient observations to be made. First, the data show that DBP and SBP confer a higher probability of T2D over the entire range of values in black individuals compared to white individuals and in females compared to males. Second, for any given level of HDL or triglycerides, black individuals have a higher probability of T2D than white individuals; however, probabilities tend to equalize at the extremes of very high HDL and very low triglyceride values. Third, probabilities appear nearly indistinguishable over the range of blood glucose, BMI, HDL, and triglyceride values when males are compared with females. When black males and females are compared with their white counterparts, the data for HDL and triglycerides also visually appear indistinguishable. Finally, the probability of incident T2D declines as a function of age; however, probabilities were higher at any given age in black individuals than white individuals and, to a lesser extent, in females than males.

Finally, Fig 4 illustrates the distribution of risk among individuals in the population as a function of race. The distribution of predicted probabilities is right-shifted towards higher risk among black individuals (mean 0.19, median 0.14) compared with white individuals (mean

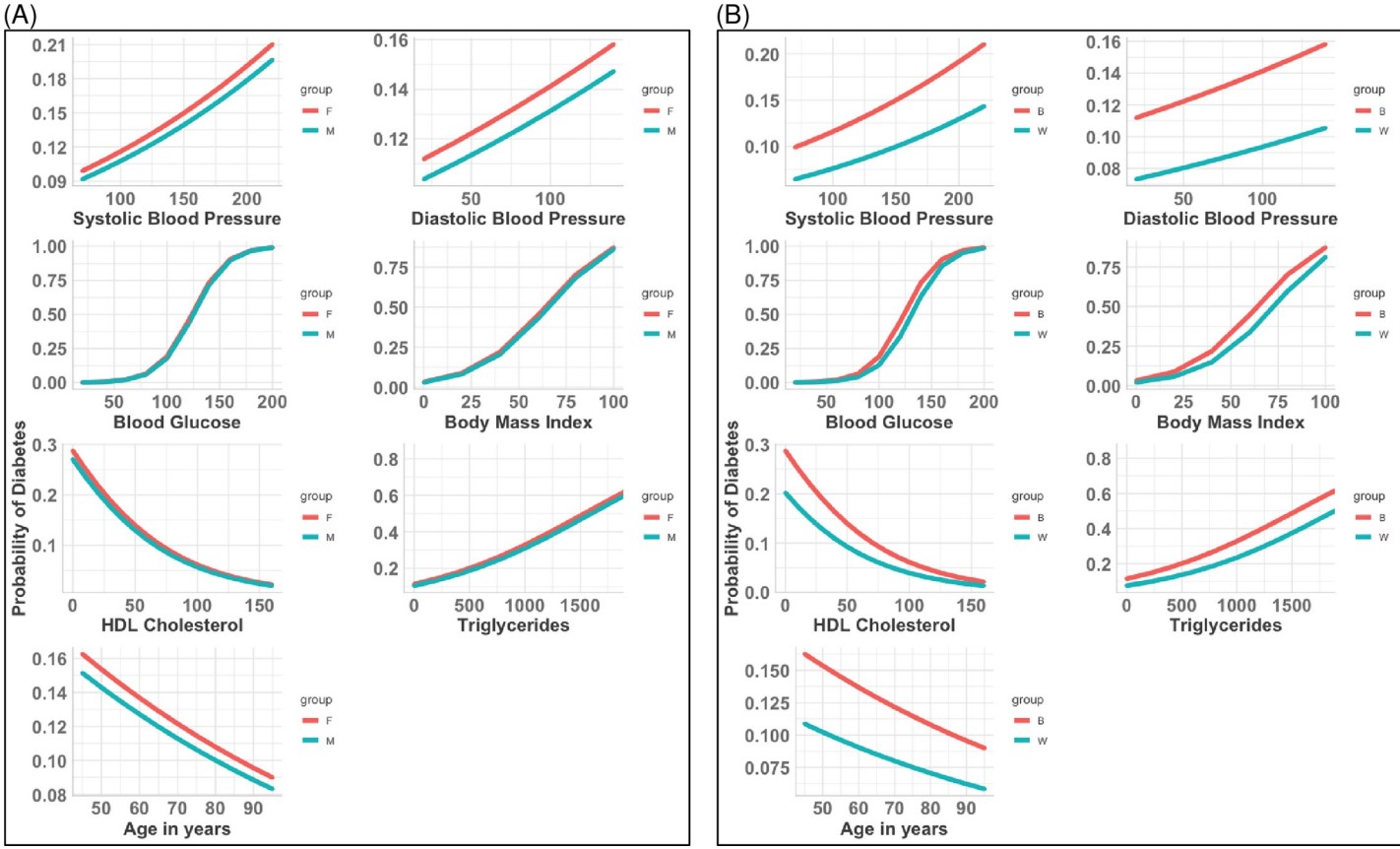

**Fig 3. Predicted probabilities for each predictor associated with type 2 diabetes by sex and race.** (A) By sex. (B) By race. B, black; F, female; HDL, high-density lipoprotein; M, male; W, white.

0.11, median 0.07). Furthermore, in both races, the validity of these predictions based on observed frequencies is quite robust over the full range of predicted probability.

## Discussion

In the current study, we present novel findings: (1) a practical and robust T2D risk calculation for incident T2D based on metabolic syndrome criteria; (2) a tool with improved capability for predicting progression to T2D compared with other commonly used paradigms (i.e., models developed in the Framingham Heart Study and by the American Diabetes Association), and generated using only quantitative data readily available to the clinician; (3) the first risk prediction tool, to our knowledge, for individuals of African descent derived from a large scientific US sample; (4) development and validation of the risk calculation model across 2 national cohorts in both black and white men and women; and (5) a unique T2D risk model that incorporates Bayesian hierarchical techniques into its risk prediction calculation. Metabolic syndrome traits constitute the basis of the prediction model, and the high AUC values highlight insulin resistance as the central pathophysiological process giving rise to these traits in the pathogenesis of T2D.

### Quantitative and qualitative difference from other scores

This study substantially advances our previous work in smaller, regional cohorts, which demonstrated that metabolic syndrome traits can be used to predict progression to T2D in

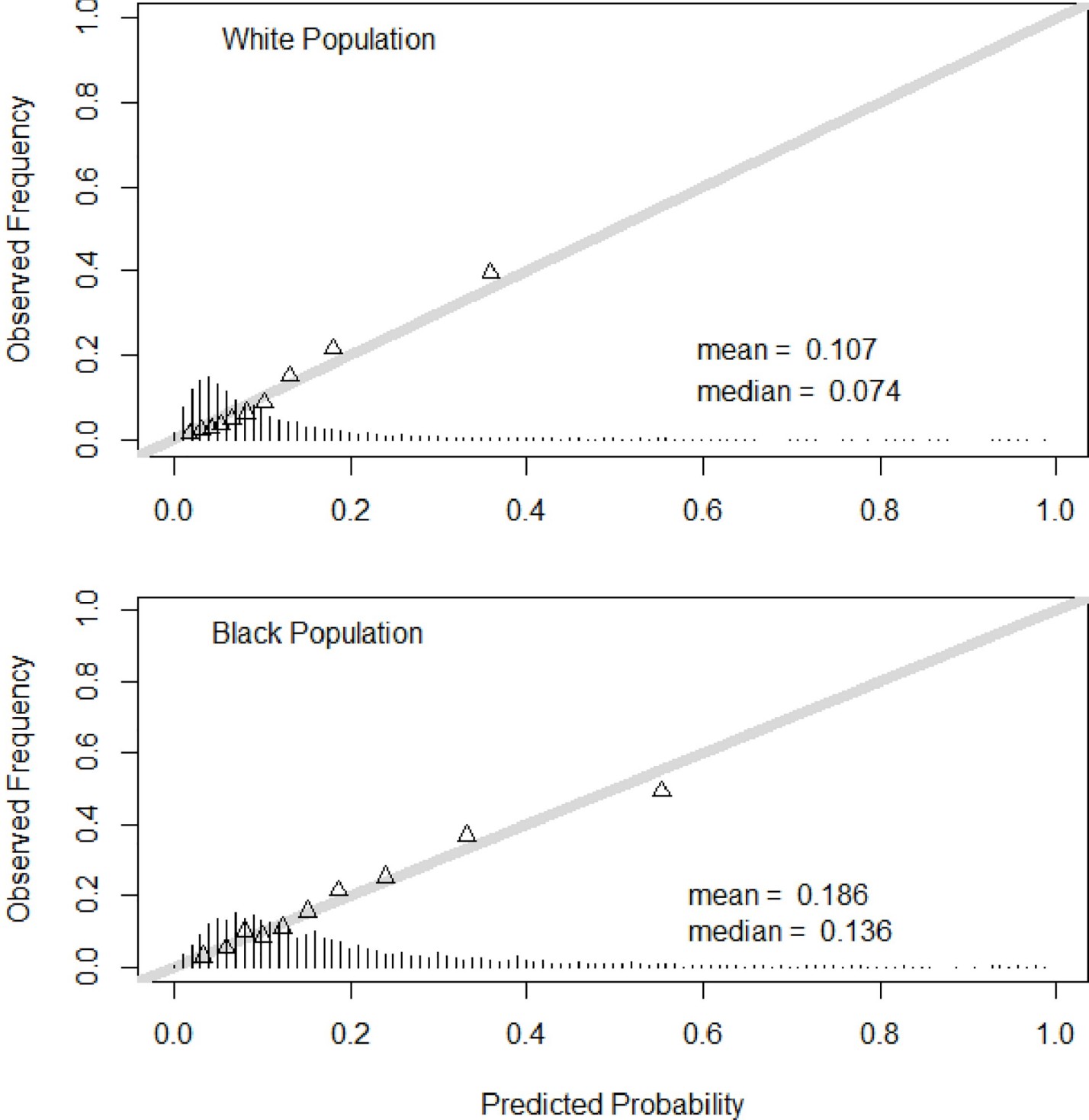

**Fig 4. Validity of predictions of incident type 2 diabetes in the development sample for white and black populations.** The distribution of predicted probabilities is shown at the bottom of the graphs. The mean and median of the predicted probability are also shown. The triangles indicate the observed frequencies by deciles of predicted probability.

individuals with overweight or obesity. An earlier iteration of the idea for a tool that used metabolic syndrome trait thresholds assigned patients to 5 discreet risk strata and assumed that each trait contributed equally to T2D risk [19]. To enhance the predictive value, the binary predictors (i.e., values above and below threshold values) were differentially weighted based on their ability to confer risk for T2D and used to generate a numerical risk score. The relative proportion of risk attributable to each trait did vary as a function of race; however, the cohorts were smaller than the REGARDS cohort and did not represent a national sample of black and white individuals. The current version is a tool created using the Bayesian logistic regression approach (implemented in BhGLM) that effectively weights the contribution to overall risk for each factor over the continuum of values and incorporates effects of race and sex. Indeed, compared with the previous iteration, for which the AUC for the ROC was 0.72 [19], the AUC was improved to 0.79 when the model was fitted using REGARDS data. Also, by using the largest black American cohort currently available for these types of studies, this tool now provides a uniquely robust quantitative risk assessment in black Americans.

## Clinical implications

The current Bayesian logistic model quantifies the 10-year risk for developing T2D. Weight loss medications and structured lifestyle interventions designed to achieve weight loss have been demonstrated to be highly effective in preventing T2D among patients with overweight or obesity [27]. Obesity, however, is highly prevalent, and weight loss interventions are laborious and entail clinical costs. Risk assessment can be used to identify those individuals at highest risk of T2D in whom weight loss interventions will have a higher benefit/risk ratio and be most cost-effective. More research is justified to assess the potential of our predictive model for individualizing care and selecting interventions to prevent cardiometabolic disease. For example, in a pooled study of 3,286 individuals who were overweight or had obesity participating in a clinical trial employing a weight loss medication (phentermine/topiramate extended release), the earlier iteration of the T2D risk model [19] effectively stratified T2D risk, and demonstrated that numbers needed to treat to prevent 1 case of T2D were markedly reduced in participants with higher risk scores at baseline [3]. Therefore, the current model offers healthcare professionals a more robust tool to assess T2D risk using quantitative clinical data that would be available based on clinical practice guidelines for patients with obesity [28].

To enhance the clinical utility of this tool, we additionally examined whether BMI could be substituted for waist circumference since waist circumference is not routinely measured in clinical venues. We found that the substitution of BMI for waist circumference did result in a minimal decrease in AUC; however, risk prediction remained robust such that BMI can be substituted for waist circumference in risk prediction.

## Strengths and limitations

The main strength of this study is the use of a large, nationally sampled, biracial cohort with validation in a second cohort. The participants are well characterized, and only reproducible quantitative data (e.g., as opposed to less reliable or subjective data such as family history or reported physical activity) are used in generating the risk score. This allowed us to create a more meaningful, interactive system using readily available clinical data, which can be applied to quantify T2D risk in individual patients. Thus, this approach has clinical utility for identifying those most likely to benefit from therapeutic interventions to prevent T2D.

A limitation of this study is that we only have 2 time points from which to assess 10-year risk of T2D; therefore, no time-to-event models were applied. Between the first and second REGARDS survey, 8,532 participants withdrew from further follow-up; upon inspecting

demographic and metabolic profile (including the 8 traits examined) differences between these 8,532 participants and those who remained, only baseline BMI and DBP showed no significant difference. While the reasons for withdrawing from follow-up are unknown, previous work in this population shows that missing data do not change exposure outcome relationships in a study such as REGARDS [29]. Participants were only non-Hispanic white or non-Hispanic black, so generalizability to other populations will require caution, and future studies that address this issue would extend the racial/ethnic reach of risk assessment using our model. The mean age of the REGARDS participants at baseline was 63.12 years, so generalizability to younger populations is not advised. We did not have physical activity or family history information, so were unable to input these when comparing this tool to the Framingham and American Diabetes Association tools.

## Conclusion

The tool presented here, using nationally sampled data from black and white Americans, has high model discrimination using readily available quantitative clinical information. The predictive value is enhanced by adding race (black or white) data. This study also quantified the differential contribution of metabolic syndrome traits to T2D risk among black and white men and women, and established the first robust tool to our knowledge for predicting T2D among individuals of African descent. Weight loss achieved by structured lifestyle interventions and anti-obesity medications is highly effective in preventing progression to T2D [30–32]. This tool can be used by clinicians and researchers to quantitatively assess T2D risk among patients with overweight/obesity. Hopefully, rational strategies for the medical care of patients with obesity based on risk will lead to greater access to evidence-based therapies.

## Supporting information

**S1 TRIPOD Checklist. Prediction model development.**
(DOCX)

**S1 Table. External evaluations for Bayesian logistic model and 4 alternative methods.**
(DOCX)

**S2 Table. Ranges of variables used to calculate T2D probabilities.**
(DOCX)

**S1 Text. Comparison with alternative methods.**
(DOCX)

**S2 Text. Calculating the predictive risk probabilities of incident T2D.**
(DOCX)

## Acknowledgments

The authors would like to thank the other investigators, the staff, and the participants of the REGARDS study for their valuable contributions.

## Author Contributions

**Conceptualization:** Lua Wilkinson, W. Timothy Garvey.

**Data curation:** Suzanne Judd.

**Formal analysis:** Lua Wilkinson, Nengjun Yi.

**Methodology:** Nengjun Yi, Tapan Mehta, W. Timothy Garvey.

**Software:** Nengjun Yi.

**Validation:** Lua Wilkinson, Nengjun Yi.

**Visualization:** Nengjun Yi, W. Timothy Garvey.

**Writing – original draft:** Lua Wilkinson.

**Writing – review & editing:** Lua Wilkinson, Nengjun Yi, Tapan Mehta, Suzanne Judd, W. Timothy Garvey.

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
