## [Editor Report · Decision Letter 0]

6 Feb 2020

Dear Dr Wilkinson, 

Thank you for submitting your manuscript entitled "Robust Prediction of Incident Diabetes Using Quantitative Clinical Data and a Bayesian Logistic Model for Cardiometabolic Disease Staging" for consideration by PLOS Medicine.

Your manuscript has now been evaluated by the PLOS Medicine editorial staff [as well as by an academic editor with relevant expertise] and I am writing to let you know that we would like to send your submission out for external peer review.

Kind regards,

Adya Misra, PhD,

Senior Editor

PLOS Medicine

---

## [Decision Letter · Decision Letter 1]

18 May 2020

Dear Dr. Wilkinson,

Thank you very much for submitting your manuscript "Robust Prediction of Incident Diabetes Using Quantitative Clinical Data and a Bayesian Logistic Model for Cardiometabolic Disease Staging" (PMEDICINE-D-20-00156R1) for consideration at PLOS Medicine. 

[LINK]

In light of these reviews, I am afraid that we will not be able to accept the manuscript for publication in the journal in its current form, but we would like to consider a revised version that addresses the reviewers' and editors' comments. Obviously we cannot make any decision about publication until we have seen the revised manuscript and your response, and we plan to seek re-review by one or more of the reviewers. 

We expect to receive your revised manuscript by May 29 2020 11:59PM. Please email us (plosmedicine@plos.org) if you have any questions or concerns.

We look forward to receiving your revised manuscript. 

Sincerely,

Adya Misra, PhD

Senior Editor 

PLOS Medicine

plosmedicine.org

Please revise your title according to PLOS Medicine's style. Your title must be nondeclarative and not a question. It should begin with main concept if possible. "Effect of" should be used only if causality can be inferred, i.e., for an RCT. Please place the study design ("A randomized controlled trial," "A retrospective study," "A modelling study," etc.) in the subtitle (ie, after a colon).

Abstract

Please provide brief participant demographics from both cohorts

If you meant T2D, please do replace instances of “diabetes” with Type 2 diabetes or T2D. 

Last sentence of the methods and findings section should be a limitation of your study design/methodology

Conclusions

Please start this section with “our results show” or similar 

Please avoid overreaching conclusions and use of adjectives such as “superior” to describe the predictive model 

Author Summary

Prospective analysis plan

Did your study have a prospective protocol or analysis plan? Please state this (either way) early in the Methods section.

Please ensure that the study is reported according to TRIPOD guideline, and include the completed checklist as Supporting Information. When completing the checklist, please use section and paragraph numbers, rather than page numbers. Please add the following statement, or similar, to the Methods: "This study is reported as per the Transparent Reporting of a Multivariable Prediction Model for Individual Prognosis or Diagnosis guideline (S1 Checklist)."

Please report your study according to the relevant guideline, which can be found here: http://www.equator-network.org/

Introduction

Could you add a reference at lines 67-68?

Could you please add a space between the text and reference brackets throughout, followed by a full stop.

Please introduce NHANES on first view 

Line 105- please introduce HDL

Methods

Line 130- please can you add United States here?

Please provide the full name of the ethics committee that approved the protocol(s)

Line 142- it would be useful to have a bit more information regarding what analyses/measurements were carried out even if brief. You may wish to add a citation in addition, if this has been previously published

Please format your bibliography to Vancouver style

Comments from the reviewers:

Reviewer #1: The stated purpose of this analysis is to create a tool using quantitative predictors available in real-world clinical practice that identifies individuals who are most likely to benefit from therapies to prevent diabetes.

Comments:

Is the REGARDS dataset (2003-2007), collected for stroke patients, representative of the wider population in order to analyse T2DM incidence?

How do the baseline population characteristics compare to the wider population, in order to extrapolate results?

"External validation was performed using 9,710 participants from Atherosclerotic Risk in Communities (ARIC) (1987-1989), observed through 1996-1998."

Is this population an unbiased sampling frame for T2DM?

"...Atherosclerotic Risk in Communities (ARIC)[11] cohorts to predict incident diabetes with specificity for sex and race. CMDS was developed using quantitative measures of metabolic syndrome traits (i.e., ATP-III criteria)[12], with the limitation that these cohorts were not designed as nationally representative."

Are the authors referring to the same cohort that they are using for their external validation? 

How did the authors cope with any potential changes in definition or testing methods of T2DM over time between baseline and follow up?

The authors use a Bayesian logistic model using full-information continuous predictors. The novelty of this research piece is in its' application of continuous variables and interactions.

The method of Bayesian logistic modelling, and the means of measuring and comparing performance, seem appropriate given the context of this research question.

"Between the first and second visit, 5,713 individuals died, and 8,532 withdrew from further follow-up, leaving a population of 15,938 with follow-up data available."

What was cause of death (i.e. were any T2DM related?)? Were reasons for withdrawing from follow up provided (in order to understand if this missing data can be considered to be missing at random)?

"Diabetes was defined by having a fasting blood glucose level ≥126mg/dL, a non-fasting blood glucose ≥ 200mg/dL, self-reported diabetes or on diabetes medication"

Is there a risk of misclassification for cases of self diagnosed T2DM? This is part of the published ARIC protocol, but what is the potential impact in this setting?

There are some grammatical errors in the text, for example the statistical methods section."Cauchy priors, which has advantage of providing"

Did the authors assess correlation between parameters in the model, and the effect this might have on the model outcome and interpretation?

The authors provide a clear description of findings in the results section, aligned with informative tables and figures.

"additional reason for greater predictive value is that the previous models did not include age.[19] "

Can the authors compare their model with previous models updated to include age (albeit dichotomously), in order to compare how much uplift in the model performance is down to the inclusion of this variable, and how much is due to the novel application of Bayesian modelling?

Reviewer #2: This is a well-conducted study that developed a Bayesian logistic model using full-information continuous predictors to predict T2DM risk. This tool can be used for diabetes prevention efforts including weight loss therapy by allowing clinicians to target high risk individuals in a manner that could be used to optimize outcomes.

The authors developed the model in REGARDS and validated it in ARIC, two large national cohorts including both White and Black men and women. They compared the Bayesian method with several other predictive modeling methods, including lasso, generalized additive model, random forests, and support vector machine learning, and found that the Bayesian logistic model outperformed these alternative approaches.

They further compared the AUC from the Bayesian logistic model to the CMDS model, the Framingham and American Diabetes Association risk scores. The AUC for the Bayesian logistic model was superior at AUC 0.79 to other models or scores, such as the CMDS model (AUC 0.75), the Framingham scoring system (AUC 0.76) and American Diabetes Association scoring system (AUC 0.64). 

Reviewer #3: This manuscript conducted a Bayesian logistic model using the full-information continuous analysis with the nationally sampled data from white and black American adults to predict T2DM risk. The manuscript is well written and provides a powerful predictive tool to used for diabetes prevention. There were some issues related to this manuscript.

1. As the authors indicated that there were 8,532 participants withdrew from further follow-up in the REGARDS. Since that the REGARDS is a nationally sampled longitudinal survey, are there any differences between these 8,532 participants and the participants remained for the further analysis in the present study?

2. Line 230, age (45-92) should be age (45-92 y).

3. In the Table A2, the min value of BMI, waist circumference, blood glucose, and HDL are extremely low. it is better to exclude these participants with outlier. Also, the unit (kg/m2) for BMI and cm for waist circumference should add in this table.

4. The mean age of the REGARDS survey is 63.12 years. Therefore, generalizability to younger population also requires caution.

[LINK]

---

## [Editor Report · Decision Letter 2]

10 Jun 2020

Dear Dr. Wilkinson,

Thank you very much for re-submitting your manuscript "Development and Validation of a Model for Predicting Incident Type-2 Diabetes Using Quantitative Clinical Data and a Bayesian Logistic Model: A Nationwide Cohort and Modelling Study" (PMEDICINE-D-20-00156R2) for review by PLOS Medicine.

I have discussed the paper with my colleagues and the academic editor and it was also seen again by reviewers. I am pleased to say that provided the remaining editorial and production issues are dealt with we are planning to accept the paper for publication in the journal.

[LINK]

We look forward to receiving the revised manuscript by Jun 15 2020 11:59PM. 

Sincerely,

Adya Misra, PhD

Senior Editor 

PLOS Medicine

plosmedicine.org

Requests from Editors:

COI – can you please say who is a paid employee of Novo Nordisk. I think this needs clarity. 

Abstract-could you add some more demographic information, like mean age and maybe BMI ranges.

Line 237 “We found that our Bayesian logistic model outperformed these alternative approaches.” Please can they provide a call out to Table A1 where the comparison is. 

Reference call outs should be in square brackets please

Line 101- suggest rephrasing “along ethnic, social, or economic lines”

Throughout- please take care to avoid saying “participants were obese” and instead say “participants suffered from obesity” or similar to avoid the use of stigmatising language. For example Line 258

Please provide exact p-values, for example at line 304 unless the p-value is <0.001 and check that the p-values are provided throughout, where appropriate 

Line 337 should say “with improved capability” 

Please temper the assertions of primacy by adding “to our knowledge” in the discussion

Please add a sentence in the methods to note the analyses were prespecified and that the analysis plans are provided as SI files

Please remove all iterations of "[Internet]" from the reference list.

I think Table 1 is not visible in the manuscript PDF and may need adjusting 

Please remove page numbers from the TRIPOD checklist as these are likely to change. Instead please use paragraphs and sections

Comments from Reviewers:

[LINK]

---

## [Editor Report · Decision Letter 3]

13 Jul 2020

Dear Dr Wilkinson, 

On behalf of my colleagues and the academic editor, Dr. Karine Clément, I am delighted to inform you that your manuscript entitled "Development and Validation of a Model for Predicting Incident Type-2 Diabetes Using Quantitative Clinical Data and a Bayesian Logistic Model: A Nationwide Cohort and Modelling Study" (PMEDICINE-D-20-00156R3) has been accepted for publication in PLOS Medicine. 

PRODUCTION PROCESS

PRESS

PROFILE INFORMATION

Thank you again for submitting the manuscript to PLOS Medicine. We look forward to publishing it. 

Best wishes, 

Adya Misra, PhD

Senior Editor 

PLOS Medicine

plosmedicine.org